# Inertial Sensor Technologies—Their Role in Equine Gait Analysis, a Review

**DOI:** 10.3390/s23146301

**Published:** 2023-07-11

**Authors:** Cristian Mihăiță Crecan, Cosmin Petru Peștean

**Affiliations:** University of Agricultural Sciences and Veterinary Medicine Cluj-Napoca, 400372 Cluj-Napoca, Romania; cosmin.pestean@usamvcluj.ro

**Keywords:** inertial measurement unit, lameness, gait analysis, horse–rider interaction, lameness detection, sedative drug

## Abstract

Objective gait analysis provides valuable information about the locomotion characteristics of sound and lame horses. Due to their high accuracy and sensitivity, inertial measurement units (IMUs) have gained popularity over objective measurement techniques such as force plates and optical motion capture (OMC) systems. IMUs are wearable sensors that measure acceleration forces and angular velocities, providing the possibility of a non-invasive and continuous monitoring of horse gait during walk, trot, or canter during field conditions. The present narrative review aimed to describe the inertial sensor technologies and summarize their role in equine gait analysis. The literature was searched using general terms related to inertial sensors and their applicability, gait analysis methods, and lameness evaluation. The efficacy and performance of IMU-based methods for the assessment of normal gait, detection of lameness, analysis of horse–rider interaction, as well as the influence of sedative drugs, are discussed and compared with force plate and OMC techniques. The collected evidence indicated that IMU-based sensor systems can monitor and quantify horse locomotion with high accuracy and precision, having comparable or superior performance to objective measurement techniques. IMUs are reliable tools for the evaluation of horse–rider interactions. The observed efficacy and performance of IMU systems in equine gait analysis warrant further research in this population, with special focus on the potential implementation of novel techniques described and validated in humans.

## 1. Introduction

In equine science, objective gait analysis methods have been frequently used to assess movement asymmetry and locomotion characteristics of horses, especially as part of lameness evaluation. Lameness has a high prevalence in horses and can have serious consequences for animals, owners, and veterinarians, including removal from athletic activity, reduced performances, and financial loss [1,2]. Lameness may also affect spinal biomechanics [3,4].

Objective asymmetry measurements by gait analysis techniques have been widely implemented; their role is to complement subjective evaluation by measuring small changes in gait patterns that cannot be captured by the human eye. Objective evaluation methods can mainly be divided into two categories: kinetic and kinematic. Kinetic methods measure the forces resulting from movement, while kinematic methods analyze the movements of body segments.

Among kinetic methods, force platforms (FPs) are the most widespread and standard instruments for objective lameness assessment. FPs measure the ground reaction force (GRF) in three dimensions, as exerted on a limb during the stance phase. Kinetic gait analysis involving GRF measurements has been used for quantification of normal gait [5,6,7] and lameness [8,9,10,11]. GRF can be assessed by methods measuring the forces produced when the hoof of the horse hits a FP mounted in/on the ground and by methods involving hoof-mounted devices, e.g., instrumented shoes [12,13,14]. FPs are accurate instruments with high sensitivity and specificity, but the data collection process is complex and time-consuming: multiple measurements of each limb are needed to collect information for a complete examination [10,15,16].

For kinematic analysis, optical motion capture (OMC) systems have been used successfully [17,18]. These systems use reflective markers attached to the body and numerous cameras distributed across the room to track the 3D position of the markers. OMC systems are precise and accurate [4], but data interpretation is complex, and the repeatability, validity, and precision of parameters differ between analyses, thus reducing the reproducibility of the measurements and relevance of data. Additionally, both FP and OMC systems are expensive and are restricted to laboratory conditions, reducing their suitability for routine diagnostic use in general practice [17,19,20].

Techniques using wireless, sensor-based inertial measurement units (IMUs) are becoming more popular as a tool for objective locomotor assessment in horses and are considered a versatile and cost-effective alternative to FP and OMC systems [21,22,23]. A system of small IMU sensors attached to different body segments of the horse (head, withers, pelvis, forelimbs, and hind limbs) allows for data collection during unrestricted movement and for investigation of gait parameters outside the laboratory [24,25]. IMU systems have found large applicability in gait event analysis to detect, localize, and quantify lameness, to objectivize results of flexion tests and/or perineural anesthesia [26,27], to evaluate of horse–rider interactions, and to set up of effective training and coaching sessions [28,29,30,31,32,33]. Even modern smartphones incorporating accelerometers and gyroscopes can be used as an equine gait analysis device to accurately detect movement symmetry [34].

Numerous studies have compared the objective lameness detection methods with subjective evaluation. Hardeman et al. [35] found that overall between- and within-veterinarian agreement was good on identifying the lame limb and measured as acceptable to poor when establishing the lameness grade. OMC data and subjective evaluations correlated well, with minor significant differences (*p* ≤ 0.003) in lameness grades between veterinarians and between assessment conditions [35]. In another study, lameness evaluation of horses with an inertial sensor system was significantly correlated with subjective examinations (*p* < 0.05) but lacked strong agreement [30]. Inter-veterinarian agreement of veterinarians in live clinical evaluations of horses trotting in a straight line was higher than in video evaluations [12]. Moderate agreement was detected between the body-mounted IMUs and highly or moderately experienced clinicians in the live clinical evaluation of hind limb lameness, while determination of lameness improvement after anesthesia had a strong association between IMU and highly experienced veterinarians [12]. All these studies suggested that sensor-based IMUs can strengthen but not replace subjective lameness assessment of horses.

IMU-based systems have been extensively characterized in relation to human motion analyses and have gained applicability in numerous fields, including posturography [36,37,38], sports [39,40,41], and clinical practice [42,43,44]. While the use of IMU systems in equine science is less documented, we hypothesize that they represent practical tools for equine clinicians and researchers alike. In this narrative review, we aimed to discuss the performance, efficacy, and applicability of IMU-based sensor systems used in equine gait analysis, based on published evidence. Publications addressing these topics were identified by performing a non-systematic literature search in PubMed, using the terms “horse locomotion”, “lameness”, “equine gait analysis”, “optical motion capture”, “force plate”, and “inertial measurement unit”. Publications that reported data considered by the authors to be relevant to the scope of this review were included, and their main findings were summarized in a descriptive way.

## 2. Inertial Measurement Units

IMUs are wearable sensors that measure acceleration forces and angular velocities and can also provide orientation estimates. These sensors allow a non-invasive and continuous monitoring of horse gait during walk, trot, or canter [45,46]. The large amount of data collected by sensor technology can be used to develop accurate gait classification models. Methods for accurate gait classification have been developed using IMU-generated data and machine learning, achieving more than 97% accuracy [46,47,48].

Several lameness detection systems are now available on the market, e.g., the Lameness Locator^®^, Equigait^®^, the Equimoves^®^ system, or Xsens^®^ (Table 1). The Lameness Locator (LL) detects and quantifies movement asymmetry based on two single-axis accelerometers attached to the head and sacrum and one single-axis gyroscope on the right forelimb, joined via wireless connection to a computer for data processing and analysis [49]. Vertical acceleration measured by LL is used to determine asymmetries in head and pelvic position between the left and right halves of the stride in trot. Due to the small number of sensors, LL can only be used in trot and provides results only for the upper body of the horse. The EquiMoves system consists of eight Promove-mini wireless IMUs connected to a computer and measures acceleration, angular velocity (gyroscope), and magnetic field intensity [28]. Our group has also described an original, sensor-based IMU system, Lameness Detector 0.1, for quantification of lameness in horses with different lameness degrees in one fore- or hind limb [25].

## 3. Locomotor Analysis in Sound Horses Using IMUs

Promove-mini sensors were extensively used to assess horse locomotion and detect gait events during walking and trot. Such sensor-based techniques have been evaluated and validated versus OMC and FP systems.

In a study conducted by Braganca et al. [50], Warmblood horses were equipped with metacarpal/metatarsal Promove-mini sensors and reflective markers for OMC and were hand walked and trotted over FP. The accuracy of and precision of hoof-on/hoof-off events were calculated from the IMU data (Table 2). The best performing IMU algorithm for stance duration at walk had accuracy, precision, and percentage of error values of 28.5 ms, 31.6 ms, and 3.7% for the forelimbs and of −5.5 ms, 20.1 ms, and −0.8% for the hind limbs, respectively. At trot, accuracy, precision and percentage of error were −27.6 ms, 8.8 ms, and −8.4% for the forelimbs and 6.3 ms, 33.5 ms, 9.1% for the hind limbs, respectively [50].

Tijssen et al. [51] studied two algorithms for detection of hoof events from the linear acceleration and angular velocity signals measured by hoof-mounted Promove-mini wireless sensors in walk and trot on a hard surface. The methods were validated versus FP data. The angular velocity method was more accurate for the hoof-on detection, while the acceleration method was superior for the hoof-off detection (Table 2) [51].

Recently, Hatrisse et al. [46] compared foot-on and heel-off events detected with Promove-mini sensors attached to the cannon bone, hooves, and withers with the method used by Tijssen et al. [51], presented above. In this study, horses were trotted on hard and soft straight lines and cycles. The method showed bias ranging between −1.94% and 3.45% of stride duration for the forelimb and hind limb hoof-on and heel-off detection and demonstrated comparable accuracy for stride detection in various conditions and surfaces other than hard surfaces (Table 2) [46].

In another study, IMU sensors were mounted on the hooves, pasterns, and cannon bones of the left forelimbs and hind limbs; the collected hoof-on/hoof-off event timing data were compared to data collected by Tijssen et al. [51]. Walk and trot strides were recorded on asphalt, grass, and sand. Pastern-based IMUs were found to be the most accurate and precise for detecting gait events, with mean errors of 1 and 6 ms, depending on the limb and gait, on asphalt (Table 2) [52].

Several studies demonstrated that the location of wearable sensors affects the accuracy of gait analysis. Steinke et al. [53] found that sensors mounted on the right forelimb vertical axis gave the best results (intra-class correlation coefficient [ICC]: 1.0; error: 6.8%, mean step count difference: 1.3), followed by sensors attached to the right hind limb and withers [53]. Similarly, in a study assessing step count accuracy and threshold acceleration values for each gait, data collected from the right front leg was more accurate and closer to the steps counted from video analysis than data from sensors located on the head or hind leg [54].

Hagen et al. [55] evaluated breakover duration using a hoof-mounted IMU sensor system and showed that trimming had no significant influence on breakover duration, while heel elevation and shoeing with a plain steel shoe increased breakover duration [55].

Methods using gyroscopic data from a single IMU mounted on the cannon bone of horses for the detection of foot-on/foot-off events were more precise than methods using accelerometric data, with a bias <0.6% of stride duration for foot-on and 0.1% for foot-off events [56]. In addition, stance and stride duration measurements had a higher precision [56]. As opposed to OMC, these angular velocity methods were also able to accurately estimate protraction and retraction angles of horse limbs [57].

MTx (Xsens^®^) IMUs mounted on the distal limbs and sacrum of the horse can also accurately and precisely detect hoof-on/hoof-off and stance events. The detected bias was −7 (standard deviation [SD]: 23) ms for front limb hoof-on, 0.7 (SD: 37) ms for hoof-off, and −0.02 (SD: 37) ms for stance. Hind limb hoof-on was detected with a bias of −4 (SD: 25) ms, hoof-off with a bias of 6 (SD: 21) ms, and stance with a bias of 0.2 (SD: 28) ms, respectively [58]. Pelvis-mounted MTx sensors were used to estimate foot contact timing from vertical velocity events. Accuracy and precision of measurements matched the data from hoof-mounted accelerometer when minimum velocity was used for walk (mean difference: 15 [SD: 18] ms across horses) and velocity zero-crossing for trot (mean difference between −4 [SD: 14] to 12 [SD: 7]) [59]. Warner et al. [23] developed and validated a method for quantifying back movement in horses, based on spine-mounted MTx sensors. The collected data showed an acceptable accuracy and consistency when compared with OMC. Dorsoventral and mediolateral displacement data were observed within ±4–5 (SD: ±2) mm of OMC [23].

## 4. Lameness Detection Using IMUs

Lameness is an alteration of the normal gait caused by structural or functional disorder in the locomotion system that manifests in gait asymmetry. Early and accurate assessment of the presence and severity of the lameness is essential to take timely and effective measures. The classical way to detect lameness consists of initial palpation of the limbs and visual evaluation of the horse at rest, followed by observation of the horse in motion at walk and trot to evaluate asymmetry in the vertical displacement of head or pelvis between consecutive steps [60]. Therefore, movement asymmetries detected by subjective visual gait examination are the most common clinical signs of lameness and are considered the most important criterion for lameness definition. Although experienced equine veterinarians have a higher inter-observer agreement in detection of gait asymmetries when compared with inexperienced observers, studies have shown that subjective evaluation is not very reliable for horses with mild to moderate lameness [61,62,63]. Between experienced clinicians grading lameness according to the American Association of Equine Practitioners’ (AAEP) scale, 91.3% agreement was reported when the mean AAEP lameness score was >1.5, and 61.9% agreement when the mean score was ≤1.5 [62].

Although its definition is still not standardized and frequently challenging, lameness is mainly considered a pathological condition. In some cases, subtle lameness and minor gait asymmetry can only be observed when the horse is ridden, performing a certain task, or moving in a certain direction. Moreover, not all asymmetries are caused by lameness [64,65], and not all individuals have the same movement symmetry [66]. Therefore, the use of IMU systems alone or in combination with subjective evaluation procedures can ensure a comprehensive analysis of horse locomotion under field conditions. Good repeatability and inter-operator reproducibility of IMU-based gait analysis systems have been demonstrated [67,68].

Bell et al. [11] compared pelvic height differences collected with IMU sensors with horizontal and vertical GRFs from FP measurements. The difference in minimum pelvic position (PD_min_) between the right and left halves of a stride was moderately associated with the difference in peak vertical GRF, but it had little association with any horizontal GRF. The difference in maximum pelvic position (PD_max_) was strongly associated with a transfer of vertical-to-horizontal ground reaction impulse in the second half of the stance but not with a difference in the peak vertical GRF [11].

In a study conducted by Pagliara et al. [69], fetlock joint angles quantified by IMUs (MOVIT system G1) demonstrated high accuracy at walk (root mean square error [RMSE] 8.23° ± 3.74°; Pearson correlation coefficient [PCC] 0.95 ± 0.03) and trot (RMSE 9.44° ± 3.96°; PCC 0.96 ± 0.02) in both sound (RMSE 7.91° ± 3.19°; PCC 0.97 ± 0.03) and lame horses (RMSE 9.78° ± 4.33°; PCC 0.95 ± 0.03) when compared to OMC. Results were in temporal agreement in 97.33% of the cases, with the mean bias between methods being around 0 [69] (Table 3). Roepstorff et al. [70] found that an IMU system (ETB Pegasus) measured a higher range of motion (ROM) of the sagittal metatarsal/metacarpal bone angles in horses at walk and trot than the OMC (bias 1.6°; precision 1.96° [SD: 1.9°) [70]. Bosch et al. [28] identified a good agreement and a low level of bias between EquiMoves system and OMC in terms of locomotor parameters of both sagittal (protraction, retraction, sagittal range of motion) and coronal (adduction, abduction, coronal range of motion) planes. The ICC results show high reproducibility of the measurements between the two systems for almost all experiments [28] (Table 3).

The Lameness Locator was compared with subjective lameness evaluations (blinded and unblinded) and FP for the detection of forelimb mild lameness induced by creating a unilateral carpal osteochondral fragment in the middle carpal joint [71]. The percentage of horses identified as lame was 60% by the Lameness Locator and 54% by subjective evaluation. These values were higher when compared to FP (40%). Blinded subjective evaluation and the Lameness Locator agreed more often on which forelimb was lame (50%) than blinded subjective assessment and FP (38% each). The best agreement was identified between subjective evaluation and the IMU system [71]. In another study, the Lameness Locator selected the lame limb after a median of 5 (range: 1–11) half turns, while three veterinarians identified the lame limb after a median of 6 (range: 2–13) half turns (*p* < 0.0001). The IMU system selected the correct limb before the veterinarians in 58.33% of cases, the evaluators selected the correct limb before the IMU sensors in 8.33% of cases [72]. In the study conducted by Keegan et al. [30], values measured by the Lameness Locator for both forelimbs and hind limbs of lame horses trotting in straight line were positively and significantly associated with subjective evaluation results (*p* < 0.05) [30]. Agreement between IMU measures and subjective assessment were fair to moderate for forelimb lameness classification and were slight to fair for hind limb lameness classification [30]. Lopes et al. [73] reported that irregular gait was identified with the Lameness Locator in 68.6% of cases. There was a significant disagreement (*p* ≤ 0.004) between sensor-based evaluation and subjective assessment; this difference was not present when the sensitivity of the IMU system was reduced by considering mild lameness not significant [73].

Flexion tests can help with localizing lameness and are often part of a routine lameness examination. Changes in pelvic movement following positive flexion tests can be detected by IMUs. The Lameness Locator has also been evaluated for assessment of the response to flexion. There was a strong correlation between measurements for pelvic movement asymmetry (PMA), and for the difference in pelvis minimum/maximum positions (PD_min_/PD_max_) measurements (*p* < 0.001). PMA (*p* = 0.021) and PD_max_ (*p* = 0.05) significantly increased following a positive flexion test [32].

Horse velocity has been demonstrated to influence both subjective and objective evaluation of lameness, especially in horses with mild lameness [74,75,76]. Moorman et al. [77] conducted a study to assess the effect of horse speed on the kinematic output from the Lameness Locator and found a significant effect on stride rate (*p* = 0.0025) and PD_min_ (*p* = 0.0234) of hind limb lameness. No significant differences on forelimb lameness kinematics (difference in head minimum/maximum positions [HD_max_, HD_min_], vector sum) were identified [77]. Another study used signals detected from the EquiMoves system in combination with machine learning to create a model for estimating the speed of horses. The model was highly accurate and able to estimate speed regardless of the IMUs location on body or gait; this model was validated for two different breeds and five gaits (walk, trot, tölt, canter, and pace) [78]. Galloping horses preserved natural asymmetry of head, pelvic, and limb motion, regardless of whether or not they had induced lameness of the forelimb or hind limb [79]. Pfau et al. [27] showed that the Mtx (Xsens) sensor system can detect mild lameness with high accuracy and precision via measuring vertical movement of the tuber coxae [27].

## 5. Horse–Rider Interaction and IMUs

IMUs are becoming an important tool for monitoring the training process of sport horses and horse–rider interactions. In a recent study, jumping and movement parameters were collected using Seaver^®^ IMU devices, and the relationships between these traits were estimated. Weak to moderate (but significant, *p* < 0.05) correlations between walk and jumping parameters were identified for 21 out of 27 possible correlations [80]. The same group of researchers has demonstrated that variability of the jumping parameters was lower for experienced horses compared with inexperienced ones; parameters that differed significantly between the two horse groups were height of jump, frequency of approach strides, acceleration of taking off, and energy by landing (*p* ≤ 0.01) [81]. Schmutz et al. [82] developed a model for estimating the horse speed per stride based on accelerometric and gyroscopic data collected by IMUs and processed using a machine learning approach. The sensor was fixed in the saddle pommel, close to the withers of the horse, and the collected data were synchronized to a four-camera tracking system to obtain reference stride speed. The proposed model was able to estimate the horse speed per stride with and average accuracy of 0.6 m/s [82]. IMUs can also be used for the evaluation of free jumping performances during regular breeding. Ricard et al. [83] used an IMU sensor (Equimetrix^®^) to estimate the correlation between the genetic component of jumping ability and the jumping performance during a breeding program. Jump duration and judgement scores were heritable and genetically correlated to competition performance: 0.59 (standard error [SE]: 0.13) for jump duration and 0.60 (SE: 0.11) −0.77 (SE: 0.12) for judgement scores [83].

The Xsens^®^ 3D motion tracking system was used to analyse horse–rider interactions. Eckardt et al. [29] quantified the horse–rider interactions in different gaits (walk, sitting trot, and canter) considering both professional and beginner riders. Horse–rider interaction was defined in terms of the time lag of cross-correlation between specific parameters of the rider and the horse: segment angles of the rider’s pelvis and trunk and the horse’s trunk. Significant differences were detected in the various gaits (*p* ≤ 0.038 for 5/6 parameters) but not in the skill levels (professional or beginner) of the riders (*p* ≥ 0.078). Cross-correlation analysis indicated a better interaction in sagittal plane than in frontal plane, regardless of gaits and skill levels [29]. When assessing the rider’s pelvis and the horse’s trunk, significant differences have been found in anterior–posterior rotations (*p* ≤ 0.05) but not in lateral rotations in walk, trot, and canter [84]. Other studies that have used the Xsens^®^ technology to study the biomechanics of the equine pelvic, thoracic, and lumbar regions have demonstrated that the saddle and the rider have a significant effect on the kinematics of the equine spine. MacKechnie-Guire et al. [85] reported changes in the thoracolumbosacral region of dressage horses when ridden in sitting trot as compared to trotting in hand. Increased range of motion (ROM) was found in the caudal thoracic and lumbar regions (T18-L3) in sitting trot. Cranial thoracic motions remained unchanged, but heading rotational values declined [85]. In another study, the ROM of the caudal thoracic and thoracolumbar regions decreased by −1.3 (0.4)° and −0.6 (0.2)°, respectively, while protraction and retraction angles increased during rising trot as compared to trotting in hand. Moreover, the rider’s effect on the kinematics of the horse spine was also different in the two phases of the rising trot [86]. Locomotion pattern modifications were detected before and after training sessions in horses. Among the kinematic indicators evaluated, indicators comparing gait cycle kinematics (Pearson’s correlation coefficients and root mean square deviations) and estimating movement smoothness (spectral arc length) indicated changes following training [87].

IMUs were also used in studies designed to allow unmanned self-coaching through classification of horse gaits and recognizing the rider’s posture for calibration. Using the results of these analyses, the rider can be self-coached in the correct motion corresponding to the classified gait and rider posture [31,88].

## 6. Tranquilizer, Sedative and Analgesic Agents and IMUs

Sedatives such as tranquilizers and analgesics are sometimes needed to perform a lameness examination in horses. However, this can change the locomotion pattern of horses, further complicating the detection of subtle lameness [89]. In the study conducted by da Silva Azevedo et al. [90], the influence of acepromazine maleate and xylazine hydrochloride on the gait pattern of horses undergoing objective lameness evaluation was assessed using the Lameness Locator. The use of a tranquilizer or sedative did not interfere with the gait and lameness of horses with different behavior and lameness intensity, as evaluated by inertial sensors [90].

Leelamankong et al. [91] assessed the effect of perineural analgesia administered to the deep branch of the lateral plantar nerve (DBLPN block) and of intra-articular analgesia administered to the tarsometatarsal joint (TMT block) on hindlimb lameness evaluation with the Lameness Locator. Horses showing lameness improvement following the DBLPN block were subjected to the TMT block. Of the 27 horses, 10 showed improvement after the TMT block and 17 horses did not show improvement, suggesting that DBLPN and TMT blocks desensitized horses differently [91]. Similarly, Rungsri et al. (2014) showed that intra-articular anesthesia of the distal interphalangeal (DIP) joint of the forelimb and perineural analgesia of the digit desensitized different regions of the forelimb [92]. Palmar digital or abaxial sesamoid nerve blocks were performed (with 2% mepivacaine) on Day 1 to ameliorate digit pain in 22 horses with forelimb lameness and DIP joint block was performed on Day 2. Among the horses with >70% improvement following palmar digital nerve blocks and abaxial sesamoid nerve blocks, 6/11 horses and 7/11 horses had positive response to the DIP joint block, respectively [92].

According Rettig et al. [93], low-dose xylazine can be used without affecting the assessment of hind limb lameness but not of the forelimb lameness with mild severity [93]. Another study evaluated whether sedation with low-dose xylazine would alter lameness amplitude as measured by the Lameness Locator. There were no significant differences in head or pelvic movement asymmetry between xylazine- and saline-treated groups. However, a few horses with forelimb lameness from the xylazine-treated group showed a large decrease in head movement asymmetry (suggesting a decrease in forelimb lameness) at 60 min following sedation. Based on these results, low-dose sedation with xylazine may be used without the concern of potential lameness-masking effects for hindlimb lameness evaluation, but caution should be used in some horses with forelimb lameness of mild severity [93]. Techniques based on accelerometry have also detected differences in the horse gait pattern in response to sedative drugs [94].

The analgesic effect of cryotherapy in an induced lameness model was evaluated by Quam et al. [95]. Horses were assigned to cryotherapy or control group, and lameness was assessed before and 5, 10, 15, 20, 30, 45, and 60 min after treatment. Lameness improved significantly in all horses 5 min after cryotherapy (*p* ≤ 0.05), but the improvement was sustained 10 min after treatment only in the cryotherapy group. One hour of cryotherapy produced a moderate, transient analgesia lasting up to 15 min. For all horses, the median vector sum was significantly increased relative to the baseline before lameness induction (*p* ≤ 0.05) [95].

## 7. Discussion and Overall Conclusions

This narrative review described numerous examples that support the versatility and usefulness of IMUs for the evaluation of equine gait asymmetries indicative of lameness. Most studies evaluated and compared several IMU methods based on accelerometer and gyrometer data and compared their findings with reference data obtained with well-established objective kinetic and kinematic methods. The collected evidence indicated that IMU-based sensor systems can monitor and quantify horse locomotion with high accuracy and precision, having comparable or superior performance to objective measurement techniques such as FPs and OMC systems.

Based on the published data reviewed here, we can conclude that IMUs can play an important role in equine medicine and research. Their high accuracy and precision, as well as their good performance has been demonstrated both in field and in laboratory conditions. As their sensitivity is superior to that of the human eye, IMUs can represent a valuable tool especially for junior veterinarians faced with the challenge of diagnosing mild lameness. Their advantages also include the possibility of gathering objective movement data in field conditions, where the use of well-established methods like FPs and OMC would be impractical or cumbersome. However, IMUs also have several disadvantages, one of which is represented by their usually high cost. To obtain reliable and reproducible data, care must be taken in placing the sensors in the right position; we strongly advise veterinarians who are considering introducing IMUs in their clinical practice to strictly adhere to the sensor placement recommended by the manufacturer of each system. An additional limitation is represented by the fact that a stable internet connection is needed to avoid errors in data.

In human studies, novel IMU-based methodologies and procedures are continuously emerging to assess movement analysis and biomechanical properties of the foot [96]. In human musculoskeletal research, a combination of IMUs, OMC, machine learning gait prediction, and finite element analysis has been used for injury monitoring, treatment, and rehabilitation [97,98]. These techniques can also be extrapolated to animal studies, with research in this direction having recently been conducted [99]. Data collected by IMU systems are valuable for the evaluation of the horse–rider interaction during dressage riding, training of horses, or coaching. Equine-assisted therapies are recognized as effective tools for physical therapy, treatment of neurodevelopmental disorders, children with attention-deficit/hyperactivity, or maintaining mental health [100,101,102]. Therefore, further characterization of horse–rider interaction is warranted. The reviewed data suggest that overall, the use of sedative agents does not interfere with lameness evaluation methods performed using IMUs, though its effect should be closely monitored as it might be dependent of dosing and chemical properties of the active molecule.

Study findings should be interpreted considering the potential limitations. The present study is a narrative review without a pre-specified search and selection methodology (as opposed to systematic reviews); hence, the conclusions drawn could be biased and could not be applicable in a more general context.

## Figures and Tables

**Table 1 sensors-23-06301-t001:** Inertial measurement units.

IMU	Number of Sensors	Sensor Position	Primary Measure	Source
Lameness Locator^®^	3	Pelvis, head, right front limb	Vertical movement of torso	https://equinosis.com/ (accessed on 3 July 2023).
Equigait^®^	3–8 *	Poll, withers, midline of the horse	Gait asymmetry, back movement, horse–rider interaction	https://equigait.co.uk/ (accessed on 3 July 2023).
Equimoves^®^	8	Poll, withers, sacrum, sternum, forelimb, hind limb	Hoof-event detection, limb angles, upper-body symmentry	https://equimoves.nl/ (accessed on 3 July 2023).
Xsens^®^	6	Poll, sacrum, limb	Gait asymmetry, limb movement	https://www.movella.com/products/xsens (accessed on 3 July 2023).

* Number of sensors according to the type of assessment (gait and symmetry, back movement, horse–rider interaction). For the correct placement of the sensors, please refer to the instructions provided by the manufacturer of each system.

**Table 2 sensors-23-06301-t002:** Studies assessing the efficacy of inertial measurement units (IMUs) for the analysis of normal gait parameters.

Study	Horse (N)	IMU (Position) vs. Validation System	Gait/limb/IMU Position	Hoof-On/Foot-On		Hoof-Off/Foot-Off		Stance Duration	
				Accuracy (ms)	Precision (ms)	Accuracy (ms)	Precision (ms)	Accuracy (ms)	Precision (ms)
Braganca, 2017 [50]	Warmblood, sound (N = 10)	Promove-mini (4 sensors on each limb [metacarpal/metatarsal]) vs. FP	Walk, forelimb	10.9 (M1)	27.2 (M1)	28.8 (M1)	26.0 (M1)	17.9 (M1)	35.7 (M1)
				−71.0 (M2)	31.1 (M2)	−45.2 (M2)	51.5 (M2)	25.8 (M2)	51.2 (M2)
				0.3 (M3)	11.5 (M3)	14.2 (M3)	31.0 (M4)	13.9 (M3)	31.5 (M3)
				−58.8 (M4)	46.1 (M4)	−40.5 (M4)	53.3 (M4)	18.3 (M4)	60.3 (M4)
			Walk, hind limb	14.1 (M1)	8.1 (M1)	−42.7 (M1)	12.9 (M1)	−56.8 (M1)	12.5 (M1)
				−18.3 (M2)	13.5 (M2)	−15.1 (M2)	21.2 (M2)	3.2 (M2)	23.6 (M2)
				2.0 (M3)	11.5 (M3)	−5.4 (M3)	14.3 (M3)	−7.4 (M3)	17.8 (M3)
				0.1 (M4	14.8 (M4)	−9.8 (M4)	26.6 (M4)	−9.8 (M4)	29.9 (M4)
			Trot, forelimb	−3.8 (M1)	23.9 (M1)	28.8 (M1)	17.5 (M1)	32.6 (M1)	28.1 (M1)
				−99.2 (M2)	58.0 (M2)	−26.8 (M2)	19.2 (M2)	72.4 (M2)	55.7 (M2)
				7.9 (M3)	6.7 (M3)	−3.7 (M3)	35.4 (M3)	−11.6 (M3)	34.6 (M3)
				−82.6 (M4)	61.8 (M4)	−19.7 (M4)	7.5 (M4)	62.9 (M4)	64.0 (M4)
			Trot, hind limb	16.3 (M1)	10.1 (M1)	17.6 (M1)	29.1 (M1)	1.3 (M1)	34.4 (M1)
				−10.8 (M2)	12.5 (M2)	−17.9 (M2)	46.7 (M2)	−7.1 (M2)	50.1 (M2)
				11.3 (M3)	9.1 (M3)	−2.3 (M3)	46.9 (M3)	−13.6 (M3)	48.9 (M3)
				11.3 (M3)	9.1 (M4)	−19.9 (M4)	32.7 (M4)	−31.2 (M4)	31.6 (M4)
Tijssen, 2020 [51]	Warmblood, sound (N = 7)	Promove-mini (2 sensors on right forelimb, right hind limb) vs. FP	Walk, forelimb	17.93 (Acc)	-	3.20 (Acc) 0.75 (AV)	-	−2.67 (Acc)	3.76 (Acc)
				11.06 (AV)	-			−1.33 (AV)	3.20 (AV)
			Walk, hind limb	23.96 (Acc)	-			−4.18 (Acc)	3.52 (Acc)
				3.55 (AV)	-			−2.88 (AV)	2.86 (AV)
			Trot, forelimb	13.77 (Acc)	-			−1.64 (Acc)	3.84 (Acc)
				2.39 (AV)	-			0.74 (AV)	4.98 (AV)
			Trot, hind limb	14.84 (Acc)	-			−2.39 (Acc)	6.18 (Acc)
				12.22 (AV)	-			−1.66 (AV)	4.52 (AV)
Hatrisse, 2022 ^a^ [46]	Sound horses (N = 7)	Promove-mini (7 sensors on withers, cannon bone, hooves) vs. Tijssen method	Trot, forelimb, straight line	0.40 (HG)	1.87 (HG)	1.40 (HG)	1.68 (HG)	1.02 (HG)	2.44 (HG)
				1.79 (SG)	1.65 (SG)	1.46 (SG)	1.80 (SG)	−0.34 (SG)	1.82 (SG)
			Trot, forelimb, left circle	0.96 (HG)	1.72 (HG)	−0.19 (HG)	1.68 (HG)	−1.16 (HG)	2.36 (HG)
				0.15 (SG)	3.74 (SG)	0.16 (SG)	1.77 (SG)	−0.28 (SG)	2.11 (SG)
			Trot, forelimb, right circle	1.12 (HG)	2.02 (HG)	0.08 (HG)	2.81 (HG)	−1.04 (HG)	3.37 (HG)
				1.14 (SG)	2.43 (SG)	0.38 (SG)	2.33 (SG)	−0.76 (SG)	3.17 (SG)
			Trot, hind limb, straight line	0.59 (HG)	1.40 (HG)	−0.25 (HG)	1.65 (HG)	−0.84 (HG)	1.69 (HG)
				0.88 (SG)	1.36 (SG)	−0.20 (SG)	1.34 (SG)	−1.09 (SG)	1.05 (SG)
			Trot, hind limb, left circle	0.75 (HG)	0.72 (HG)	−0.95 (HG)	1.69 (HG)	−1.71 (HG)	1.85 (HG)
				−0.56 (SG)	0.50 (SG)	−1.94 (SG)	1.17 (SG)	−1.38 (SG)	1.25 (SG)
			Trot, hind limb, right circle	3.45 (HG)	1.09 (HG)	0.24 (HG)	3.54 (HG)	−3.22 (HG)	3.63 (HG)
				−0.12 (SG)	0.73 (SG)	−0.74 (SG)	2.13 (SG)	−0.62 (SG)	2.38 (SG)
Briggs, 2021 ^b^ [52]	Sound horses (N = 11)	Shimmer3 (6 sensors on hooves, pastern, cannon bone) vs. Tijssen method	Walk, forelimb, pastern	−5 (M1)	11 (M1)	3 (M1)	16 (M1)	0.92 (asphalt) 0.52 (grass) 1.19 (sand)	2.98 (asphalt) 2.14 (grass) 5.29 (sand)
				−4 (M2)	14 (M2)	−7 (M2)	27 (M2)		
			Walk, forelimb, cannon	−5 (M1)	21 (M1)	1 (M1)	19 (M1)		
				−30 (M3)	40 (M3)	−15 (M3)	65 (M3)		
				14 (M4)	72 (M4)	−35 (M4)	117 (M4)		
			Walk, hind limb, pastern	2 (M1)	10 (M1)	6 (M1)	14 (M1)	0.10 (asphalt) −0.33 (grass) 0.22 (sand	3.42 (asphalt) 7.42 (grass) 3.60 (sand)
				−1 (M2)	10 (M2)	−3 (M2)	38 (M2)		
			Walk, hind limb, cannon	−5 (M1)	11 (M1)	−1 (M1)	15 (M1)		
				−31 (M3)	55 (M3)	−57 (M3)	69 (M3)		
				161 (M4)	412 (M4)	−15 (M4)	180 (M4)		
			Trot, forelimb, pastern	−4 (M1)	10 (M1)	−18 (M1)	23 (M1)	1.09 (asphalt) −0.57 (grass) −1.09 (sand	8.68 (asphalt) 6.67 (grass) 7.68 (sand)
				−2 (M2)	9 (M2)	4 (M2)	34 (M2)		
			Trot, forelimb, cannon	−9 (M1)	23 (M1)	−15 (M1)	33 (M1)		
				−26 (M3)	21 (M3)	62 (M3)	114 (M3)		
				43 (M4)	67 (M4)	77 (M4)	102 (M4)		
			Trot, hind limb, pastern	1 (M1)	12 (M1)	2 (M1)	9 (M1)	0.50 (asphalt) 0.92 (grass) 1.16 (sand	2.38 (asphalt) 2.26 (grass) 4.28 (sand)
				−1 (M2)	19 (M2)	15 (M2)	21 (M2)		
			Trot, hind limb, pastern	−3 (M1)	16 (M1)	−7 (M1)	14 (M1)		
				−4 (M3)	27 (M3)	−144 (M3)	62 (M3)		
				98 (M4)	147 (M4)	−72 (M4)	100 (M4)		

M, method—indicates the different methods and algorithms developed by the authors to assess the performance of the studied IMU systems; Acc, acceleration; AV, angular velocity; HG, hard ground; SG, soft ground. ^a^ Results are expressed as percentage of stride duration (%). ^b^ Stance duration expressed as percentage of stride duration (%).

**Table 3 sensors-23-06301-t003:** Studies assessing the efficacy of inertial measurement units for the evaluation of lameness.

Study	Horse (N)	IMU (Position) vs. Validation System	Parameters	RMSE ± SD (°)	PCC ± SD (°)	ICC	ROM
Pagliara, 2022 [69]	Sound and lame horses (N = 14)	MOVIT system (8 sensors on dorsal metacarpus/metatarsus and pastern) vs. OMC	Walk, sound horses	7.77 ± 3.42	0.96 ± 0.03		
			Walk, lame horses	8.68 ± 4.03	0.95 ± 0.04		
			Trot, sound horses	8.06 ± 2.99	0.97 ± 0.02		
			Trot, lame horses	10.06 ± 4.39	0.96 ± 0.03		
Bosch, 2018 [28]	Warmblood, sound (N = 7)	EquiMoves (8 sensors on poll, withers, sacrum, sternum, and limbs) vs. OMC	Walk, sagittal plane	0.67 (forelimb)	-	0.99 (retraction)	0.99
				0.67 (hind limb)	-	0.98 (protraction)	
			Walk, coronal plane	1.27 (forelimb)	-	0.92 (adduction)	0.92
				1.01 (hind limb)	-	0.92 (abduction)	
			Trot, sagittal plane	1.05 (forelimb)	-	0.97 (retraction)	0.98
				0.95 (hind limb)	-	0.97 (protraction)	
			Trot, coronal plane	1.79 (forelimb)	-	0.90 (adduction)	0.94
				1.21 (hind limb)	-	0.94 (abduction)	
Keegan, 2011 [67]	Sound and lame horses (N = 236)	IMU (3 sensors on head, pelvis, pastern), two consecutive measurements	HMA (%)	16.8 ^a^		0.885	
			HD_min_ (mm)	3.15 ^a^		0.936	
			HD_max_ (mm)	3.17 ^a^		0.900	
			PMA (%)	14.1 ^a^		0.952	
			PD_min_ (mm)	1.36 ^a^		0.935	
			PD_max_ (mm)	1.69 ^a^		0.925	
Marshall, 2012 [32]	Sound horses (N = 17)	Lameness Locator (3 sensors on head, pelvis, right forelimb), two consecutive measurements	PMA (left hind limb)		0.98		
			PMA (right hind limb)		0.94		
			PD_min_ (mm)		0.98		
			PD_max_ (mm)		0.97		

RMSE, root mean square error; PCC, Pearson correlation coefficient; ICC, intra-class correlation coefficient; ROM, range of motion; HMA, head movement asymmetry; HD_min_/HD_max_, difference in head minimum/maximum positions between right and left portions of the stride; PMA, pelvic movement asymmetry; PD_min_/PD_max_, difference in pelvis minimum/maximum positions between right and left portions of the stride. ^a^ Typical error between measurement 1 and measurement 2.

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
