# Peer review of "Inertial Sensor Technologies—Their Role in Equine Gait Analysis, a Review"

_sensors, 2023, doi:10.3390/s23146301_

Round 1

Reviewer 1 Report

Congratulations to the authors. I appreciate all the work that went into this revision. This is a carefully prepared and well written manuscript describing the use of inertial sensor technology for the evaluation of horse locomotion. The revision is well written but some English proofreading is needed.

There are a few additions or changes and minor revisions that need to be addressed.

Abstract:

- Line 12: Change “suck” by “such”

1. Introduction

- Line 64-66: This sentence can be reinforced with the paper of Hardeman et al. “A first exploration of perceived pros and cons of quantitative gait analysis in equine clinical practice” (Equine Vet Ed).

- Line 70: Regarding applicability in lameness evaluation is important to note the use of IMUs to objectivize results of flexion tests and/or perineural or intrasionovial anesthesias frequently used in routinely lameness exams in horses.

- Line 83: Change “ISUs” by “IMUs”

3. Horse Locomotor Analysis using Inertial Measurement Units

In the heading of this section, change “Inertial Measurement Units” by “IMUs”

- Line 130: Change “hoot-on” by “hoof-on”

- Line 149: Change “canon” by “cannon”

- Line 165: Please rephrase “Data showed acceptable accuracy and good…”

4. Lameness Detection using IMUs

- Lines 172-182: I think is important to mention that actually is some controversy regarding the relation between asymmetry and lameness. Asymmetry is not a clinical concept. Not all asymmetries refer to a lameness problem and not all the individuals have the same symmetry levels. For example young animals are more asymmetric than mature individuals and a dressage animal is more symmetrical than a racing thoroughbred. Kallerud et al. Equine Vet J 2021 “Objectively measured movement asymmetry in yearling Standardbred trotters” is a good example.

- Line 212: Change “lameness” by “lame”

- Line 240-242: - Line 165: Please rephrase

At the end of this section I would include all the information provided in the section 6 related to perineural analgesia, blocks etc. (Lines 305-318) since this drugs are not sedatives and are used in lameness exams.

6. Tranquilizer and Sedative Agents and IMUs

I would change this heading adding analgesic Agents (Tranquilizer, Sedative and Analgesic Agents and IMUs) since you mention at the end the evaluation of cryotherapy as an analgesic agent.

Also in this section, some publications with information regarding the use of accelerometry to evaluate the effects of sedative drugs on the horse locomotion and specifically for lameness evaluation  are lacking and I think deserve a mention.

- Line 303: Add “was evaluated” after “Locator”

7. Conclusions

- Line 343: Change “agent” by “agents”

- Line 343-344. This conclusion is not supported by some publications included in the revision. As mentioned by Rettig “Low-dose sedation with xylazine may be 325 used without the concern of potential lameness-masking effects for hindlimb lameness 326 evaluation, but caution should be used in some horses with forelimb lameness of mild severity”. Also the above mentioned papers describing an influence of sedative drugs in the locomotor pattern dependent on the dose and the molecule used are not supporting this conclusion.

English is not my mother language but I think that some English proofreading is needed. I picked some typographic errors in the text and specify them to the authors.

Author Response

General comment: Congratulations to the authors. I appreciate all the work that went into this revision. This is a carefully prepared and well written manuscript describing the use of inertial sensor technology for the evaluation of horse locomotion. The revision is well written but some English proofreading is needed. There are a few additions or changes and minor revisions that need to be addressed

  1. Abstract:

- Line 12: Change “suck” by “such”

Answer: The typo was corrected. Thank you for pointing out.

  1. Introduction

- Line 64-66: This sentence can be reinforced with the paper of Hardeman et al. “A first exploration of perceived pros and cons of quantitative gait analysis in equine clinical practice” (Equine Vet Ed).

Answer: The statement was revised and publication cited as suggested.

- Line 70: Regarding applicability in lameness evaluation is important to note the use of IMUs to objectivize results of flexion tests and/or perineural or intrasionovial anesthesias frequently used in routinely lameness exams in horses.

Answer: The corresponding paragraph was slightly amended and completed as per suggestion.

- Line 83: Change “ISUs” by “IMUs”

Answer: The typo was corrected. Thank you!

  1. Horse Locomotor Analysis using Inertial Measurement Units

In the heading of this section, change “Inertial Measurement Units” by “IMUs”

Answer: The title was changed as suggested.

- Line 130: Change “hoot-on” by “hoof-on”

Answer: The typo was corrected.

- Line 149: Change “canon” by “cannon”

Answer: The typo was corrected.

- Line 165: Please rephrase “Data showed acceptable accuracy and good…”

Answer: The statement was revised to clarify.

  1. Lameness Detection using IMUs

- Lines 172-182: I think is important to mention that actually is some controversy regarding the relation between asymmetry and lameness. Asymmetry is not a clinical concept. Not all asymmetries refer to a lameness problem and not all the individuals have the same symmetry levels. For example young animals are more asymmetric than mature individuals and a dressage animal is more symmetrical than a racing thoroughbred. Kallerud et al. Equine Vet J 2021 “Objectively measured movement asymmetry in yearling Standardbred trotters” is a good example.

Answer: Paragraph was completed as per suggestion

- Line 212: Change “lameness” by “lame”

Answer: Corrected as suggested.

- Line 240-242: - Line 165: Please rephrase

Answer: Statement corrected for clarity.

At the end of this section I would include all the information provided in the section 6 related to perineural analgesia, blocks etc. (Lines 305-318) since this drugs are not sedatives and are used in lameness exams.

  1. Tranquilizer and Sedative Agents and IMUs

I would change this heading adding analgesic Agents (Tranquilizer, Sedative and Analgesic Agents and IMUs) since you mention at the end the evaluation of cryotherapy as an analgesic agent.

Answer: The section title was revised as suggested.

Also in this section, some publications with information regarding the use of accelerometry to evaluate the effects of sedative drugs on the horse locomotion and specifically for lameness evaluation  are lacking and I think deserve a mention.

Answer: Mentioned as requested.

- Line 303: Add “was evaluated” after “Locator”

Answer: Sentence completed as suggested.

  1. Conclusions

- Line 343: Change “agent” by “agents”

Answer: Typo was corrected as requested.

- Line 343-344. This conclusion is not supported by some publications included in the revision. As mentioned by Rettig “Low-dose sedation with xylazine may be 325 used without the concern of potential lameness-masking effects for hindlimb lameness 326 evaluation, but caution should be used in some horses with forelimb lameness of mild severity”. Also the above mentioned papers describing an influence of sedative drugs in the locomotor pattern dependent on the dose and the molecule used are not supporting this conclusion.

Answer: Indeed, that publication reported negative results for forelimb lameness of mild severity. However, the majority of studies indicated minimal or no effect of sedative drugs on lameness assessment. In response to reviewer’s comments, we amended the corresponding statement in the manuscript.

Reviewer 2 Report

This is an interesting review of the inertial sensor technologies for equine gait analysis. The authors have reviewed different studies on IMU to assess the walking stages such as foot on, foot off, and stance phases. Further, they have enlisted the studies that investigated the effectiveness of IMU sensors for a lameness evaluation. Overall, the review study is fair, timely, and relevant to the scope of the journal. However, the contributions are less comprehensive and misplaced. The following concerns need to be addressed before publication:

1. The Abstract can not be presented in the form of an introduction. The authors are suggested to highlight their contributions from the present work. What are key observations the authors found at the end of this review, that need to be mentioned?

2. In the Introduction section, the background should be updated with recent and relevant works on sensing methods for equine gait analysis. The authors are suggested to highlight the motivation behind conducting this review study. What are the research questions?

3. The authors are suggested to provide a few schematic representations of the existing approaches from cited literature. It is hard to interpret the sensing layout of equipment from Table 1 and Table 2.

4. The authors should be consistent to follow the referencing styles such as Schmutz et al. (2020) and [68] are two different styles. Please revise the complete manuscript in view of the same. The statement "Significant differences were detected... skill levels of the riders" on page 10 (line 237-274), need to be expanded for quantitative findings. There are several such missing explanations in the text that need to be corrected.

5. The authors strongly suggested highlighting the research tool (such as PRISMA) to collect the relevant studies for this review. What are the inclusion and exclusion criteria?

6. There should be a separate section on the discussion of the topic that reveals the limitations of this study and possible opportunities for the readers. The key message from the conclusion is missing. The authors mentioned, "Significant differences were detected ... ... riding, training, or coaching", the sentence is misleading for the concluding remarks. 

The manuscript should be checked for average English language, punctuation errors, and grammatical mistakes. 

Author Response

General comment: This is an interesting review of the inertial sensor technologies for equine gait analysis. The authors have reviewed different studies on IMU to assess the walking stages such as foot on, foot off, and stance phases. Further, they have enlisted the studies that investigated the effectiveness of IMU sensors for a lameness evaluation. Overall, the review study is fair, timely, and relevant to the scope of the journal. However, the contributions are less comprehensive and misplaced. The following concerns need to be addressed before publication:

  1. The Abstract can not be presented in the form of an introduction. The authors are suggested to highlight their contributions from the present work. What are key observations the authors found at the end of this review, that need to be mentioned?

Answer:

  1. In the Introduction section, the background should be updated with recent and relevant works on sensing methods for equine gait analysis. The authors are suggested to highlight the motivation behind conducting this review study. What are the research questions?

Answer: We completed the introduction section with additional justification on the objective of this narrative review.

Answer:

  1. The authors should be consistent to follow the referencing styles such as Schmutz et al. (2020) and [68] are two different styles. Please revise the complete manuscript in view of the same. The statement "Significant differences were detected... skill levels of the riders" on page 10 (line 237-274), need to be expanded for quantitative findings. There are several such missing explanations in the text that need to be corrected.

Answer: We are using these two styles in two different contexts. The numbers in square bracket are used at the end of the statements, this is the style required for in-text citations. The name of the first author and publication date is used in the main text to identify/introduce the study.

  1. The authors strongly suggested highlighting the research tool (such as PRISMA) to collect the relevant studies for this review. What are the inclusion and exclusion criteria?

Answer: We did not conduct a systematic literature search, it was a narrative review based in search using terms that we considered relevant for this topic. We now completed the introduction section to better explain the motivation of this study.

  1. There should be a separate section on the discussion of the topic that reveals the limitations of this study and possible opportunities for the readers. The key message from the conclusion is missing. The authors mentioned, "Significant differences were detected ... ... riding, training, or coaching", the sentence is misleading for the concluding remarks.

Answer:  We amended the Conclusion section and completed with further information.

Reviewer 3 Report

                The paper covers very interesting information and is very trendy in the subject. The structure of the paper and the description of the material need however revision. Nowadays reviews need a description of the methodology and how the cited papers were selected – you need to give the engine details (eg. Google Scholar, Web of Science), and inform on the keywords that were used (eg. IMU, ..). The total number of papers found and the way of selection if used.  That part should be named material. The high-quality review paper put also the hypothesis that is rejected or confirmed (eg. IMU is a precise tool in horse riding).

               It would be also good to underline somewhere that IMUs in horses are/were used first for lameness detection and that there are more papers on it (check it in numbers), otherwise, it is difficult to understand why more than half of your work is on lameness.       The structure needs to be reconstructed – as the introduction is on lameness, not on IMUS (at least to L64). The title is good as far as someone knows that the IMUs were used mostly for lameness detection.  Perhaps the “with special attendance to lameness evaluation” could be more clear. The part about lameness may be good but as the shortened paragraph in the lameness section. You write about IMUs and that should be the main subject of the introduction, not the lameness definition. So the introduction should be corrected, as well as the second part – from L90 should be the main introduction part.   It is also not clear when you write once about % of changes than on ms. Please try to compare it even by calculating mean values in the discussion part.

In detail:

Give the year of publication in L135,

Exclude on board in L 178,

Avoid using shortcuts in reviews - L 228, the especially first time it should be given in detail, or give at least a shortcut list

L 345 – please correct all editorial information – left your info or withdraw it.

Author Response

General comment: The paper covers very interesting information and is very trendy in the subject. The structure of the paper and the description of the material need however revision. Nowadays reviews need a description of the methodology and how the cited papers were selected – you need to give the engine details (eg. Google Scholar, Web of Science), and inform on the keywords that were used (eg. IMU, ..). The total number of papers found and the way of selection if used. That part should be named material. The high-quality review paper put also the hypothesis that is rejected or confirmed (eg. IMU is a precise tool in horse riding).

Answer: This is a narrative review rather than a systematic one, however, we completed the introduction section to better describe the search strategy.

It would be also good to underline somewhere that IMUs in horses are/were used first for lameness detection and that there are more papers on it (check it in numbers), otherwise, it is difficult to understand why more than half of your work is on lameness.       The structure needs to be reconstructed – as the introduction is on lameness, not on IMUS (at least to L64). The title is good as far as someone knows that the IMUs were used mostly for lameness detection.  Perhaps the “with special attendance to lameness evaluation” could be more clear. The part about lameness may be good but as the shortened paragraph in the lameness section. You write about IMUs and that should be the main subject of the introduction, not the lameness definition. So the introduction should be corrected, as well as the second part – from L90 should be the main introduction part. It is also not clear when you write once about % of changes than on ms. Please try to compare it even by calculating mean values in the discussion part.

Answer: To address reviewer’s query, we rearranged the introduction section focusing more on gait analysis instead of lameness.

In detail:

Give the year of publication in L135,

Answer: Added as requested

Exclude on board in L 178,

Answer: Amended accordingly

Avoid using shortcuts in reviews - L 228, the especially first time it should be given in detail, or give at least a shortcut list

Answer: Abbreviations defined at first appearance as requested

L 345 – please correct all editorial information – left your info or withdraw it.

Reviewer 4 Report

In this study, the authors sought to describe the inertial sensor technologies and summarize their role in equine gait analysis. The purposes of this study appear to be specific, but the research novelty and structure should be further highlighted and improved. The authors need to make major revisions to the manuscript. Here below are specific suggestions for this study.

1.     Title: This shall be a systematic scoping review. For a review, generally a very focused question is needed. This review is more focusing on the scope of the discipline.

2.     Abstract, background, please briefly explain the novelty of this review in one or two sentences.

3.     Abstract, the reviewer suggests that this session should be rewritten based on the logical structure of the scientific study, which basically includes background, purpose, methods, results, and conclusions.

4.     Introduction, I could not see the gap that the authors were going to bridge, and neither can I see the significance of this study properly. What is the current research status in this field? Are there any similar systematic review studies? If so, what are the differences or advantages of this article compared to the previous research? If not, what is the significance/novelty of this article?

5.     At the end of this section, the author stated that sensor-based IMUs can strengthen but not replace subjective lameness assessment of horses, how can it be strengthened specifically? In addition, please revise the argument to be more neutral if the authors were not recognized it as a research gap.

6.     It is suggested that the authors must consider making a throughout review of previous studies and further highlight the importance of conducting this review. What are the advantages of this study? What can this study add to this topic? These questions should be clearly explained in this part.

7.     “2. Inertial Measurement Units”, I suggest the author use a table in this section to present more clearly the types of lameness detection systems and their corresponding functions on the current market for comparison.

8.     Sections 3 to 6, for the core of the review, the paragraph included too much specifics and this turns the review into a list instead of a critical review. Please remove some unnecessary details, particularly those settings already mentioned in the Tables. Please only retains those details if the article had a very special or interesting point.

9.     Table 1: "M, method, M1, M2, M3, M4" could not be understood.

10.  Please add a discussion session and my suggestion was on two aspects. The first aspect is about the main findings of this review and what are the current research gap. The second aspect is the future research direction, for example, this part can cover how we can apply comprehensive methods to monitor, treat, and rehabilitate horses with lameness. Currently, in relevant human musculoskeletal research, a combination of IMU, motion capture, machine learning, and finite element analysis has been used for injury monitoring, treatment, and rehabilitation, involving machine learning gait prediction, finite element motion simulation, and so on. These research contents and methods can be extrapolated to animal studies, and research in this direction has now already been conducted. Some references that could be cited are listed below, including machine learning and finite element analysis.

[1]    The influence of running shoe with different carbon-fiber plate designs on internal foot mechanics: A pilot computational analysis. Journal of Biomechanics, 2023, 111597. Human finite element analysis.

[2]    Machine learning approach to diabetic foot risk classification with biomechanics data. Gait & Posture, 2022, 97, 30-31. Foot risk classification machine learning.

[3]    EquiSim: An Open-Source Articulatable Statistical Model of the Equine Distal Limb. Frontiers in Veterinary Science, 2021, 8, 623318. Equine distal limb model.

11.  The conclusions should be further strengthened based on the main findings of this study.

 Moderate editing of English language

Author Response

General comment: In this study, the authors sought to describe the inertial sensor technologies and summarize their role in equine gait analysis. The purposes of this study appear to be specific, but the research novelty and structure should be further highlighted and improved. The authors need to make major revisions to the manuscript. Here below are specific suggestions for this study.

  1. Title: This shall be a systematic scoping review. For a review, generally a very focused question is needed. This review is more focusing on the scope of the discipline.

Answer: This is not a systematic literature review, it is a narrative review conducted to summarize the available information on the use of IMU systems in equine science. We completed the introduction section to better explain the rationale behind study conduct.

  1. Abstract, background, please briefly explain the novelty of this review in one or two sentences.

Answer: We revised the abstract as requested.

  1. Abstract, the reviewer suggests that this session should be rewritten based on the logical structure of the scientific study, which basically includes background, purpose, methods, results, and conclusions.

Answer: Being a narrative review, this manuscript does not have a standard structure seen at original articles or systematic reviews. However, we agree with your query and revised the abstract to follow that structure as much as possible.

  1. Introduction, I could not see the gap that the authors were going to bridge, and neither can I see the significance of this study properly. What is the current research status in this field? Are there any similar systematic review studies? If so, what are the differences or advantages of this article compared to the previous research? If not, what is the significance/novelty of this article?

Answer: IMU-based techniques are well described in human research and there new methods are continuously emerging. While there are numerous equine studies assessing the potential of these systems, less focus has been directed toward equine science and as far as we know, there are no publications reviewing the status of this research. Thus, as a first attempt, we intended to review the current literature to summarize the evidence related to IMU systems and their applicability and use in horse locomotion assessment.

  1. At the end of this section, the author stated that sensor-based IMUs can strengthen but not replace subjective lameness assessment of horses, how can it be strengthened specifically? In addition, please revise the argument to be more neutral if the authors were not recognized it as a research gap.

Answer:

  1. It is suggested that the authors must consider making a throughout review of previous studies and further highlight the importance of conducting this review. What are the advantages of this study? What can this study add to this topic? These questions should be clearly explained in this part.

Answer: We performed several searches using different combinations of search terms, however, we did not identify any review (systematic or narrative) that summarizes the publicly available information on IMUs in equine movement research. We completed the introduction section to better contextualize the rationale.

  1. “2. Inertial Measurement Units”, I suggest the author use a table in this section to present more clearly the types of lameness detection systems and their corresponding functions on the current market for comparison.

Answer: In response to reviewer query, we added a table as Table 1 to the manuscript

  1. Sections 3 to 6, for the core of the review, the paragraph included too much specifics and this turns the review into a list instead of a critical review. Please remove some unnecessary details, particularly those settings already mentioned in the Tables. Please only retains those details if the article had a very special or interesting point.

Answer: We revised the manuscript considering suggestions from all authors. In the main text, we only included data that are not presented in tables to avoid redundancy.

  1. Table 1: "M, method, M1, M2, M3, M4" could not be understood.

Answer: In these studies, authors developed several methods and algorithms to assess the performances of the studied IMU system. We completed the footnote to better explain this.

  1. Please add a discussion session and my suggestion was on two aspects. The first aspect is about the main findings of this review and what are the current research gap. The second aspect is the future research direction, for example, this part can cover how we can apply comprehensive methods to monitor, treat, and rehabilitate horses with lameness. Currently, in relevant human musculoskeletal research, a combination of IMU, motion capture, machine learning, and finite element analysis has been used for injury monitoring, treatment, and rehabilitation, involving machine learning gait prediction, finite element motion simulation, and so on. These research contents and methods can be extrapolated to animal studies, and research in this direction has now already been conducted. Some references that could be cited are listed below, including machine learning and finite element analysis.

 [1] The influence of running shoe with different carbon-fiber plate designs on internal foot mechanics: A pilot computational analysis. Journal of Biomechanics, 2023, 111597. Human finite element analysis.

[2] Machine learning approach to diabetic foot risk classification with biomechanics data. Gait & Posture, 2022, 97, 30-31. Foot risk classification machine learning.

[3] EquiSim: An Open-Source Articulatable Statistical Model of the Equine Distal Limb. Frontiers in Veterinary Science, 2021, 8, 623318. Equine distal limb model.

Answer: The Conclusion section was revised and completed with additional discussions.

  1. The conclusions should be further strengthened based on the main findings of this study.

Answer: The conclusion part of the manuscript is revised

Round 2

Reviewer 2 Report

Although the authors have addressed some of the concerns; however, the responses and revisions are insufficient to improve the quality of the manuscript. The authors did not consider the answers to the reviewer's comments seriously. The manuscript could not be accepted due to the less comprehension of the results and discussions. The two referencing styles can not be used in the same article. The authors have not responded to anything for the second part of Comment 4. The research questions are still not presented in the revised manuscript. 

Author Response

Response: In an attempt to address the reviewer’s concerns, we revisited the manuscript and made additional changes. The referencing style was uniformized, and p values were added to support claims of statistical significance (where available). The aim of the review and the method used to retrieve and analyze the included publications were also formulated more clearly. In addition, edits were made to the Results section to improve clarity, and the Discussion section was further expanded.

With regards to the following comment (received during the first round of review): “The authors are suggested to provide a few schematic representations of the existing approaches from cited literature. It is hard to interpret the sensing layout of equipment from Table 1 and Table 2.”, we apologize for omitting to provide an answer previously. While we agree that visual representation/schematics could help orient readers in where the sensors should be placed, we believe that correct placement, in accordance with the manufacturer’s instructions, is crucial to minimize potential errors in data. Therefore, as the review was not focused on methodological aspects of IMUs, we chose not to provide extensive indications/visuals on where sensors should be placed for each system, but instead to stress the importance of correct placement and direct interested readers to the manufacturer’s instructions (in the text and in the footnote of Table 1)

Reviewer 3 Report

The Authors did not correct the paper according to the remarks given earlier. They also did not answer all remarks.  The paper is not corrected in a sufficient way. Only part of the remarks was taken into account. The introduction was corrected in part. The discussion was not changed, and the methodology seems not clear enough as it is still not clear what “relevant publications” mean. The study should be repeatable – so the description of the way to choose the papers for the review is important. Some info on the number of publications obtained and checked could be useful. Narrative reviews are not scientific enough, even interestingly written. 

The discussion is not deep enough – see earlier remarks about data comparison – as well as the part of the discussion on the positive and negative sides of IMU (L 408-412), should be more detailed.

 It is really a pity that Authors publish so much good work (102 cited papers with summaries in tables)  in such an insufficient way. Please correct the manuscript according to all remarks carefully.

Author Response

Response: Narrative reviews are a recognized form of summarizing current knowledge on a topic, and, unlike systematic literature reviews, they do not require a pre-specified search strategy and inclusion/exclusion selection criteria. However, to enhance the repeatability of our search, we added additional details on the search and the selection processes used to collect the publications included in the review.

In the Discussion section, we added a paragraph with a summary of the advantages and disadvantages of IMUs.

Regarding a comment received during the previous review round: “It is also not clear when you write once about % of changes than on ms. Please try to compare it even by calculating mean values in the discussion part.”: as now stated in the description of the study objectives, our review summarized the included publications in a descriptive way, meaning data were not transformed or modified, and no meta-analysis of the results was aimed for. While we acknowledge that such an analysis would undoubtably be a valuable contribution to the body of evidence on IMU use in equine science, we believe that the type of descriptive summary provided by us can also help interested readers in gaining an overview of the current knowledge in this area. 

Reviewer 4 Report

All my questions have been well addressed, now, I recommend to accept now. 

Minor editing of English language required

Author Response

Response: The manuscript was revisited and edits were made for gramatical correctness and clarity. We thank the reviewer for the input.

Round 3

Reviewer 2 Report

As this time, the authors have adequately addressed all the concerns raised by the reviewer, the manuscript could be accepted without further revisions.